# Interprofessional Therapeutic Drug Monitoring of Carbapenems Improves ICU Care and Guideline Adherence in Acute-on-Chronic Liver Failure

**DOI:** 10.3390/antibiotics12121730

**Published:** 2023-12-14

**Authors:** Stephan Schmid, Chiara Koch, Katharina Zimmermann, Jonas Buttenschoen, Alexander Mehrl, Vlad Pavel, Sophie Schlosser-Hupf, Daniel Fleischmann, Alexander Krohn, Tobias Schilling, Martina Müller, Alexander Kratzer

**Affiliations:** 1Department of Internal Medicine I, Gastroenterology, Hepatology, Endocrinology, Rheumatology, and Infectious Diseases, University Hospital Regensburg, 93053 Regensburg, Germany; chiara.koch@ukr.de (C.K.); katharina.zimmermann@ukr.de (K.Z.); jonas.buttenschoen@ukr.de (J.B.); alexander.mehrl@ukr.de (A.M.); vlad.pavel@ukr.de (V.P.); sophie.schlosser-hupf@ukr.de (S.S.-H.); martina.mueller-schilling@ukr.de (M.M.); 2Hospital Pharmacy, University Hospital Regensburg, 93053 Regensburg, Germany; daniel.fleischmann@ukr.de (D.F.); alexander.kratzer@ukr.de (A.K.); 3Department of Interdisciplinary Acute, Emergency and Intensive Care Medicine (DIANI), Klinikum Stuttgart, 70174 Stuttgart, Germany; a.krohn@klinikum-stuttgart.de (A.K.); t.schilling@klinikum-stuttgart.de (T.S.)

**Keywords:** therapeutic drug monitoring, global health, carbapenems, intensive care unit, interprofessional collaboration, acute-on-chronic liver failure

## Abstract

(1) Background: Acute-on-chronic liver failure (ACLF) is a severe, rapidly progressing disease in patients with liver cirrhosis. Meropenem is crucial for treating severe infections. Therapeutic drug monitoring (TDM) offers an effective means to control drug dosages, especially vital for bactericidal antibiotics like meropenem. We aimed to assess the outcomes of implementing TDM for meropenem using an innovative interprofessional approach in ACLF patients on a medical intensive care unit (ICU). (2) Methods: The retrospective study was conducted on a medical ICU. The outcomes of an interprofessional approach comprising physicians, hospital pharmacists, and staff nurses to TDM for meropenem in critically ill patients with ACLF were examined in 25 patients. Meropenem was administered continuously via an infusion pump after the application of an initial loading dose. TDM was performed weekly using high-performance liquid chromatography (HPLC). Meropenem serum levels, implementation of the recommendations of the interprofessional team, and meropenem consumption were analyzed. (3) Results: Initial TDM for meropenem showed a mean meropenem serum concentration of 20.9 ± 9.6 mg/L in the 25 analyzed patients. Of note, in the initial TDM, only 16.0% of the patients had meropenem serum concentrations within the respective target range, while 84.0% exceeded this range. Follow-up TDM showed serum concentrations of 15.2 ± 5.7 mg/L (9.0–24.6) in Week 2 and 11.9 ± 2.3 mg/L (10.2–13.5) in Week 3. In Week 2, 41.7% of the patients had meropenem serum concentrations that were within the respective target range, while 58.3% of the patients were above this range. In Week 3, 50% of the analyzed serum concentrations of meropenem were within the targeted range, and 50% were above the range. In total, 100% of the advice given by the interprofessional team regarding meropenem dosing or a change in antibiotic therapy was implemented. During the intervention period, the meropenem application density was 37.9 recommended daily doses (RDD)/100 patient days (PD), compared to 42.1 RDD/100 PD in the control period, representing a 10.0% decrease. (4) Conclusions: Our interprofessional approach to TDM significantly reduced meropenem dosing, with all the team’s recommendations being implemented. This method not only improved patient safety but also considerably decreased the application density of meropenem.

## 1. Introduction

Acute-on-chronic liver failure (ACLF) is a life-threatening condition that occurs in patients with liver cirrhosis. Without liver transplantation, the 28-day mortality rate of patients with ACLF Grade 1 ranges from 18 to 25%, and in those with ACLF Grade 3 from 68 to 89% [1]. For these patients, early and comprehensive interprofessional care on the intensive care unit (ICU) is vital [2]. ACLF is mainly triggered by bacterial infections, with bacterial translocation as an important pathomechanism [3]. This is aggravated by the reduced mucus layer and a destabilization of the intestinal epithelial barrier in patients with liver cirrhosis [4]. Gram-negative bacteria are of particular importance. Infections with multidrug-resistant (MDR) bacteria have been rising in recent years [5]. Given the rapid progression of ACLF, it is crucial to promptly and effectively treat its primary trigger: bacterial infections [6]. The guidelines of the European Society for the Study of the Liver (EASL) recommend, among others, meropenem for antibiotic therapy in ACLF [7,8].

Meropenem, a highly effective beta-lactam antibiotic of the carbapenem class, is crucial for treating severe infections in patients with life-threatening diseases, especially when infections with multidrug-resistant (MDR) bacteria are suspected. It acts as a bactericidal, time-dependent drug that inhibits bacterial cell wall synthesis [9].

The plasma concentration of meropenem should exceed the minimum inhibitory concentration (MIC) of the detected or assumed pathogen for the whole dosing interval in severe infections, while optimal pathogen clearance has been demonstrated at plasma concentrations at 4–5 times exceeding the MIC [10]. Rational use of carbapenems is of paramount importance for global health, especially in light of the increasing bacterial resistance and the slow development of new antibiotics [11]. An essential aspect of the rational use of meropenem and the use of carbapenems as a class of antibiotics, in general, is the correct dosing to avoid the development of bacterial resistance and to ensure the rapid and effective action of the antibiotic. The latter is of particular importance for critically ill patients with ACLF [12,13,14,15].

Therapeutic drug monitoring (TDM) represents an effective tool for controlling the dosage of drugs [16]. In 2020, the Infection Section of the European Society of Intensive Care Medicine (ESICM) issued a position paper in which the worldwide importance of TDM in critically ill patients is emphasized [17]. Of high relevance for critically ill patients on the intensive care units are pharmacokinetic changes, significantly altered drug clearance, and altered volume distribution [18]. In ACLF, these factors are highly relevant since ACLF is accompanied by extrahepatic organ failure. In accordance, renal failure is of utmost importance, as most antibiotics, including meropenem, are primarily renally eliminated. Likewise, hemodialysis, if necessary, also plays an important role [19]. Furthermore, pharmacodynamic changes in critically ill patients on the ICUs are of relevance as pharmacodynamics links pharmacokinetic exposure with the ability of antibiotics to kill or inhibit the further growth of bacteria [20]. Recently, studies have analyzed the significance of TDM for carbapenems, including continuous vs. intermittent meropenem administration in critically ill patients [21]. However, none of these studies have focused on an interprofessional approach for patients with severe liver diseases, with ACLF being the most life-threatening condition.

In addition to the pure technical implementation of TDM, an interprofessional approach to TDM is of utmost importance [22]. This process begins with identifying patients for whom TDM should be performed. It continues through pre-analytics and analytics and is particularly focused on the interpretation and appropriate clinical response to the TDM measurements of antibiotics in individual patients. Physicians, hospital pharmacists, and staff nurses are particularly important in this interprofessional team.

Interprofessional collaboration and education are of vital importance in times of an increasing shortage of healthcare experts across the globe, increasingly complex treatment procedures in an aging population, and pandemics [23,24]. This has also been impressively demonstrated by the World Health Organization (WHO), which emphasized in its publication “Framework for Action on Interprofessional Education & Collaborative Practice” that “The World Health Organization (WHO) and its partners recognize interprofessional collaboration in education and practice as an innovative strategy that will play an important role in mitigating the global health workforce crisis” [25].

Our study aimed to investigate the outcomes of implementing TDM for meropenem in ACLF patients in the ICU using an innovative interprofessional approach. We examined the recommendations of the interprofessional team, changes in meropenem dosing, and its application density. This is the first study to explore an interprofessional approach to TDM for meropenem in ICU patients with ACLF.

## 2. Materials and Methods

### 2.1. Study Design and Patient Characteristics

This retrospective study from the Department of Internal Medicine I at the University Hospital Regensburg, Germany, evaluated the outcomes of an interprofessional approach to TDM for meropenem in ACLF patients in the ICU from October 2022 to June 2023. This was compared to a seasonally adapted control period before initiating TDM for meropenem from October 2021 to June 2022. The study was approved by the Ethics Committee at the University Hospital Regensburg with the registration number 23-3508-104. The included 25 patients were hospitalized on a medical ICU specialized in gastroenterology, hepatology, infectious diseases, endocrinology, rheumatology, and liver transplantation. This ICU, a referral center in southern Germany, provides tertiary clinical care for approximately 2.0 million people.

### 2.2. Characteristics of Interprofessional Collaboration and Education

On the ICU, an interprofessional spirit has been cultivated for several years through various projects, including the “I’M A-STAR” initiative. The “I’M A-STAR” (which stands for “Intensiv Medizinische AusbildungsSTAtion Regensburg” in German, translating to “Intensive Care Training Ward Regensburg”) project was established in 2020 in response to the SARS-CoV-2 pandemic. Here, medical students, pharmacy students, and nurse practitioner trainees collaborate as an interprofessional team under expert supervision directly at the patient’s bedside, receiving comprehensive interprofessional training. This interprofessional approach to education and work, which has already been adopted as part of the antibiotic stewardship on our ICU [26], was now extended to TDM for meropenem in patients with ACLF.

### 2.3. Application of Meropenem

Patients with ACLF who were treated on the ICU and received meropenem were discussed by the interprofessional team of physicians, pharmacists, nurses, and students in daily TDM ward rounds. The initial 1 g loading dosage of meropenem was administered via short infusion; consecutively, meropenem was administered continuously via an infusion pump. For this purpose, 1g of meropenem (Meropenem Hikma 1000 mg, Hikma Farmacêutica, Terrugem, Portugal) as a dry substance was dissolved in 50 mL of NaCl (NaCl 0.9%, B. Braun Melsungen, Melsungen, Germany) and placed in a 50 mL syringe (Original Perfusor^®^ Syringe, B. Braun Melsungen, Melsungen, Germany). Finally, an extension set (BD Pressure Rated Extension Sets, BD Switzerland, Eysins, Switzerland) was adapted onto the syringe. The carbapenem syringe was changed every 8 h to ensure stability. Thus, meropenem was administered continuously during the entire period of the analysis after an initial loading dose of 1 g.

### 2.4. Therapeutic Drug Monitoring (TDM) of Meropenem

TDM of meropenem was performed weekly, and the corresponding blood collection was performed at 8 am under steady-state conditions. Steady-state was assumed if the time interval between the start of the therapy and the first blood sampling for TDM was >24 h. For patients requiring medication with meropenem for more than seven days, follow-up TDM was performed on a weekly basis (2nd and 3rd TDM), as organ functions may change during hospitalization on the ICU.

The blood samples were transported in a cooled condition to the hospital’s central laboratory (Institute of Clinical Chemistry and Laboratory Medicine, University of Regensburg, Regensburg, Germany) where the analysis was performed using high-performance liquid chromatography (HPLC, Chromsystems, Gräfelfing, Germany). The TDM results were immediately analyzed interprofessionally upon availability.

The interprofessional team provided recommendations for meropenem dosing adjustments. Any changes to meropenem dosing, switching to another antibiotic, or discontinuation of antibiotic therapy were executed by the attending intensive care physician and documented accordingly.

### 2.5. Characterization of Patients with Acute-on-Chronic Liver Failure (ACLF)

Acute-on-chronic liver failure was defined according to the EASL CLIF guidelines [7]. The Child score [27], SOFA score [28], MELD score [29,30], and CLIF-C ACLF score [31] were calculated.

The MELD score, a well-established indicator of the mortality of patients with end-stage liver disease, was individually calculated for every patient using the following equation [29,30]:

MELD score = 9.57 × ln(serum creatinine) + 3.78 ln(total bilirubin) + 11.2 × ln(international normalized ratio) + 6.43

The CLIF-C ACLF score was calculated according to the following formula, wherein CLIF-C OF score was raised according to [31]:

CLIF-C ACLF score = 10 × (0.33 × CLIF-C-OFs + 0.04 × Age + 0.63 × ln (WBC count in 10^3^/μL) − 2)

### 2.6. Acquisition of Data for Meropenem Consumption at the ICU

The consumption of meropenem was analyzed in recommended daily doses (RDD) per 100 patient days (PD), which is an established measurement of hospital antibiotic use. According to the literature, RDD better reflects the real consumption of antibiotics than defined daily doses (DDD) defined by the World Health Organization (WHO)/Anatomical Therapeutic Chemical (ATC) classification [32,33]. To calculate the application density for the intervention period (Q4/2022–Q2/2023) and the control period (Q4/2021–Q2/2022), the amounts of meropenem used in the ward in the respective period were converted into daily doses and related to the PDs using standardization to 100 PD in accordance with the literature [34]. Data on the amount of meropenem used were provided by the pharmacy, and patient days were provided by the hospital administration.

### 2.7. Statistical Analyses and Collection of Primary Data

For this scientific analysis, primary data were obtained from the SAP^®^ (Systemanalyse Programmentwicklung, Walldorf, Germany) hospital system and the Mevavision^®^ patient data management system. Pharmacoeconomic data were provided by the hospital pharmacists. Statistical analyses were performed with the help of SPSS^®^ (Statistical Package für Social Sciences, IBM, Armonk, New York, NY, USA). A one-tailed *t*-test was performed, and *p*-values less than or equal to 0.05 were considered statistically significant.

## 3. Results

### 3.1. Patient Cohort

The demographic and clinical characteristics of the 25 study patients are presented in Table 1 and Figure 1. In total, 8 patients were female, and 17 patients were male. The age of the cohort ranged from 20 to 74 years (mean 55.4 ± 13.90 years). All patients were treated on the ICU due to ACLF.

The leading cause of liver cirrhosis was alcohol-related (n = 10, 40%). Other causes for liver cirrhosis were cryptogenic (n = 4, 16.0%), non-alcoholic fatty liver disease (n = 2, 8.0%), secondary sclerosing cholangitis (n = 3, 12.0%), autoimmune (n = 2, 8.0%), primary biliary cholangitis (n = 2, 8.0%), chronic Hepatitis B (n = 1, 4.0%), and genetic (n = 1, 4.0%). The patients with liver cirrhosis were classified according to the Child–Pugh classification: 1 (=4.0%) patient had liver cirrhosis Child–Pugh A, 5 (=20.0%) patients had liver cirrhosis Child–Pugh B, and 19 (=76.0%) patients had liver cirrhosis Child–Pugh C. The mean MELD score was 28.2 ± 9.3 (14–40).

In 21 patients (84.0%), ACLF was triggered by infections. Pneumonia was the most common underlying infection, found in eight patients (32.0%). Other infections were as follows: spontaneous bacterial peritonitis in five patients (20.0%), cholangitis in two patients (8.0%), and urinary tract infection and endocarditis each in one patient (4.0% for each), while the focus remained cryptogenic in four patients (16.0%). In four patients (16.0%), ACLF was triggered by esophageal variceal hemorrhage. All of the studied patients had been previously hospitalized in regular wards or other ICUs and had received prior antibiotic therapy. The mean SOFA score was 15.8 ± 3.6 (range: 9–22), and the mean CLIF-C-ACLF score was 60 ± 9.3 (range: 43–76). Six patients (24.0%) had ACLF Grade 1, three patients (12.0%) had ACLF Grade 2, and sixteen patients (64.0%) had ACLF Grade 3.

Mechanical ventilation was required in 24 (=96.0%) patients with ACLF, and renal replacement therapy in 14 (56.0%) patients. In total, 10 patients (=40.0%) survived, and 15 patients (=60.0%) were deceased. In the deceased patients, ACLF resulted, despite maximum intensive care therapy, in septic multiorgan failure, coagulation failure with subsequent bleeding, and circulatory failure.

The studied patient collective was critically ill, largely because enrollment occurred in an ICU and due to the complexity and high mortality associated with ACLF.

### 3.2. Initial Continuous Meropenem Dosing

The initial continuous dosing of meropenem was determined by the ICU physician based on national and international guidelines. Patients with ACLF have an expanded third compartment, which may necessitate higher doses of meropenem compared to other patients with similar kidney function (19). In 18 patients (=72.0%), the initial dosing of meropenem was 125 mg/h (=3 g/24 h). In two (=8.0%) patients, the dose was 84 mg/h (=2 g/24 h) and in five (=24.0%) patients it was 62.5 mg/h (=1.5 g/24 h).

### 3.3. Results of Initial TDM for Meropenem

Initial therapeutic drug monitoring (TDM) for meropenem showed a mean meropenem serum concentration of 20.9 ± 9.6 mg/L (8.4–39.0) in the 25 analyzed patients (Table 2, Figure 2, Appendix A). A large variability of meropenem serum concentrations was observed (range: 8.3–39.0 mg/L).

For unidentified bacterial pathogens, a meropenem serum concentration of 10 mg/L, equivalent to five times the minimal inhibitory concentration (MIC) of Pseudomonas aeruginosa, was targeted. For patients with a known causative bacterial pathogen, a meropenem serum concentration of five times the respective MIC was aimed for.

In the patients examined in this study, bacterial pathogens were rarely identified, likely due to prior antibiotic therapies. The identified bacteria included Streptococcus pneumoniae, Citrobacter species, and Proteus mirabilis. The mean MIC for meropenem was 0.125 mg/L.

The target meropenem serum concentration included +/−25% of the respective target serum concentration, i.e., a range from 7.5 mg/L to 12.5 mg/L meropenem serum concentration for non-identified bacterial pathogens. Of note, only 4 (=16.0%) patients had meropenem serum concentrations that were within the respective target range, while 21 (=84.0%) patients were outside this range, which were all above the targeted serum concentration.

These results underscore the urgent need for TDM of meropenem, especially during the initial stages of treatment.

### 3.4. Results of Follow-Up TDM for Meropenem

Follow-up TDM for patients requiring medication with meropenem for more than seven days was performed on a weekly basis. In Week 2, TDM for meropenem was performed in 12 patients, and in Week 3, TDM for meropenem was performed in 2 patients. The mean meropenem serum concentration was 15.2 ± 5.7 mg/L (9.0–24.6) in Week 2, and 11.9 ± 2.3 mg/L (10.2–13.5) in Week 3 (Table 2, Figure 2, Appendix A).

Meropenem serum concentrations decreased significantly over time, as observed in the second and third measurements. In Week 2, five (=41.7%) patients had serum concentrations within the target range, while seven (=58.3%) patients were above this range; none were below. By Week 3, one analyzed serum concentration of meropenem was within the targeted range, and one was above this range. Consequently, the instances of meropenem serum concentrations falling outside the targeted range decreased over time.

### 3.5. Recommendations of the Interprofessional Team concerning the Use of Meropenem in the Context of TDM and Implementation of These Recommendations

All meropenem serum concentrations determined by HPLC at the hospital’s central laboratory were discussed by the interprofessional team of physicians, hospital pharmacists, and nurses, as well as medical students, pharmacy students, and nurse practitioner trainees during interprofessional grand rounds on the same day as the results were obtained. When advising on any dose adjustment of meropenem or change in antibiotic therapy, the patient’s overall clinical presentation and the microbiological situation were also considered.

Based on the primary measurement of meropenem serum concentration, the interprofessional team suggested a decrease in meropenem dosage in 10 (40.0%) patients and a change in antibiotic therapy, primarily to piperacillin/tazobactam, in 5 (20.0%) patients. No increase in meropenem dosage was suggested; in 10 (=40.0%) patients, the interprofessional team recommended no adjustment of meropenem dosage or a change in antibiotic therapy (Table 3, Appendix A). Regarding the analysis of the results of the second meropenem dosing, the interprofessional team recommended a decrease in meropenem dosage in one (=8.3%) patient and a change in antibiotic therapy, primarily to piperacillin/tazobactam, in three (=25.0%) patients. In seven (58.3%) patients, no adjustment of meropenem dosage or change in antibiotic therapy was suggested: an increase in meropenem dosage was never suggested. No changes in meropenem dosage or change in antibiotic therapy were suggested by the interprofessional team during the analysis of the third serum concentration of meropenem.

Analyzing all recommendations of the interprofessional team regarding antibiotics, 60.6% of the recommendations were related to TDM, highlighting the importance of TDM. Of note, 100% of the advice regarding meropenem dosing in the context of TDM or a change in antibiotic therapy given by the interprofessional team was implemented.

### 3.6. Analysis of Meropenem Consumption at the ICU

For the analyses regarding total meropenem consumption at the ICU, the previous year’s period was used for comparison, in which the TDM of meropenem had not yet been introduced. To allow a good comparison, seasonality was considered. The application density of meropenem was calculated in recommended daily doses (RDD) per 100 patient days (PD), which is an established measurement of hospital antibiotic use. Encouragingly, there was a significant drop in meropenem application density from 49.9 RDD/100 PD (Quarter(Q) 4/2021), 39.4 RDD/100 PD (Q1/2022), and 37.7 RDD/100 PD (Q2/2022) in the control period to 46.0 RDD/100 PD (Q4/2022), 33.1 RDD/100 PD (Q1/2023), and 34.5 RDD/100 PD (Q2/2023) in the intervention period. Therefore, the mean meropenem application density in the intervention period was 37.9 RDD/100 PD, while the meropenem application density in the control period was 42.1 RDD/100 PD, resulting in a significant 10.0% decrease (*p* = 0.02) in meropenem application density (Table 4, Figure 3). Thus, TDM is an important aid in the rational use of carbapenems, which is vital for global health.

## 4. Discussion

This is the first study analyzing an interprofessional approach to TDM in patients with ACLF. Our study shows that therapeutic drug monitoring (TDM) is crucial for an optimized meropenem dosage in critically ill patients with ACLF. Based on the interprofessional team’s recommendations, the meropenem dosage was reduced in 40% of all cases during the initial TDM assessment. Therefore, our interprofessional team approach to TDM resulted in a reduced application density of meropenem on the ICU. All recommendations made by the interprofessional team were implemented in 100% of cases.

Previous studies on meropenem TDM in ACLF were significant but did not explore interprofessional aspects or meropenem application density changes [19,20].

### 4.1. Cirrhosis-Associated Immune Dysfunction (CAID) and ACLF

Patients with liver cirrhosis are particularly susceptible to infections due to cirrhosis-associated immune dysfunction (CAID). The intensity of CAID correlates with the severity of liver diseases. Infections, again, are the most common causes of ACLF, with MDR bacteria becoming increasingly important. ACLF is characterized by severe inflammation and immune paralysis [35]. For these patients with ACLF, urgent and individualized care on an ICU is crucial, which, in our view and based on the data presented, should include TDM with an interprofessional team approach.

### 4.2. Continuous Administration of Meropenem

In our study, meropenem was given continuously during the entire period of the analysis after an initial loading dose. Continuous administration of meropenem might increase the efficacy of meropenem in critically ill patients. This is supported by numerous studies that have demonstrated a higher clinical improvement rate and lower mortality rate when meropenem was administered continuously, as opposed to intermittent administration [36,37,38,39]. Recently, in July 2023, however, Monti et al. could not find an improved composite outcome in the “The Mercy Randomized Clinical Trial” when meropenem was given continuously in critically ill patients [21]. It is worth noting that “The MERCY Randomized Clinical Trial” and other studies did not perform TDM of meropenem, which might contribute to the inconsistent data. Furthermore, a significant subgroup, the immunocompromised patients, were omitted in “The MERCY Randomized Clinical Trial”.

### 4.3. TDM for Meropenem

We think that TDM is crucial when administrating meropenem continuously to ensure the correct dosage, in particular for immunocompromised patients like those with ACLF, especially as meropenem is a bactericidal antibiotic. Thus, the dosing of meropenem is particularly critical [9]. In our study, meropenem was administered continuously controlled by interprofessional TDM according to current recommendations [17], in contrast to an earlier study by Grensemann et al., where meropenem was not administered continuously [19].

TDM for meropenem in our study revealed that only 16.0% of initial serum concentrations of meropenem were within the targeted range when meropenem was administered continuously over 24 h. In total, 84.0% of all meropenem serum concentrations were above the target range, and no meropenem serum concentration was below the targeted range. Meropenem is excreted primarily by the kidney and is in 20–25% metabolized through extrarenal mechanisms. In severe renal insufficiency, which is very common in patients with ACLF, extrarenal metabolism can increase up to 50% [40]. This results in the potential need to adjust the meropenem dosage according to liver function, e.g., using the MELD score [20].

In our study, the interprofessional therapeutic drug monitoring (TDM) revealed that 60% of initial analyses were associated with meropenem overdosing in intensive care patients. Both underdosing and overdosing present critical risks. A reduction in the dose or a change to another antibiotic was recommended by the interprofessional team when analyzing the initial TDM results in 60.0%. Of note, the interprofessional team always took the exact clinical presentation of the patient into account and did not recommend a reduction in the dosage of meropenem in highly septic patients (24.0%). Personalized and adequate dosing of meropenem can substantially reduce the side effect profile for the individual patient. This is of even greater importance in ACLF as patients with ACLF have a significantly higher vulnerability to the side effects of meropenem. Typical side effects of meropenem include seizures and Clostridioides difficile infections (CDI) [41,42]. A statistically significant correlation between elevated plasma levels of meropenem and neurotoxicity has been shown in the literature [43]. Beumier et al. summarize that a plasma concentration of 16 mg/L for meropenem, which means exceeding the MIC for pseudomonas aeruginosa by eight times, exposes patients to avoidable neurotoxicity without clinical benefit in treating the underlying infection [43]. In the setting of ACLF, patients are considerably more susceptible to seizures due to the lowered seizure threshold because of hepatic encephalopathy and delirium and to C. difficile infections due to the severity of dysbiosis that is often present [4,44,45].

### 4.4. Interprofessional Collaboration and Shared Decision-Making

The recommendations from the interprofessional team, comprising physicians, pharmacists, and nurses, were fully 100% implemented. Much lower implementation rates are known from studies conducted in the context of antibiotic stewardship (ABS), which were as low as 50% [46,47]. By involving physicians, pharmacists, and staff nurses in decision-making, the team on the ICU fully implemented and supported the proposed changes of the interprofessional team.

Interprofessional education and collaboration, as well as shared decision-making, are becoming increasingly important in times of global challenges and pandemics [48,49,50]. Interprofessional collaboration is crucial in the highly complex environment of critical care medicine [25]. As the significance of interprofessional collaboration grows in the future, we have established the concept of the “I’M A-STAR” (=Intensiv Medizinische AusbildungsSTAtion Regensburg = German for “Intensive care training ward Regensburg”) project in 2020 as a response to the SARS-CoV2 pandemic to introduce nursing trainees, as well as medical and pharmaceutical students, to interprofessional cooperation at an early stage of their education. As part of this training, we have involved the medical and pharmaceutical students, as well as the nursing trainees, very closely in the interprofessional TDM for meropenem in ACLF. In addition to an increase in knowledge among students and nurses, this has led to a greater acceptance of TDM among the team and has significantly increased awareness of this vital issue. The high implementation rate of the appropriate advice regarding antibiotic adjustments in the interprofessional team is undoubtedly due to the close involvement of the I’M A-STAR project in TDM.

### 4.5. Impact of TDM of Meropenem on Local Application Density

Our interprofessional approach to TDM of meropenem in ACLF has significantly reduced the application density of meropenem from 42.1 RDD/100 PD in the control period to 37.9 RDD/100 PD in the intervention period. Meropenem is a crucial antibiotic for treating MDR bacteria [51]. Responsible and rational use of carbapenems is essential to avoid resistance, e.g., by carbapenemase-forming bacteria [11,52]. We suggest to implement interprofessional TDM based on the clinical benefit shown in this study.

### 4.6. Limitations

This study is a retrospective single-center study.

## 5. Conclusions

This study showed that our interprofessional TDM approach for meropenem not only optimized dosing but also achieved complete adherence to team recommendations, enhancing patient safety and decreasing meropenem use. The rational use of meropenem is vital for both individual patients and global health. Furthermore, our data show the critical importance of interprofessional collaboration to ensure the quality of care during global health workforce challenges.

## Figures and Tables

**Figure 1 antibiotics-12-01730-f001:**
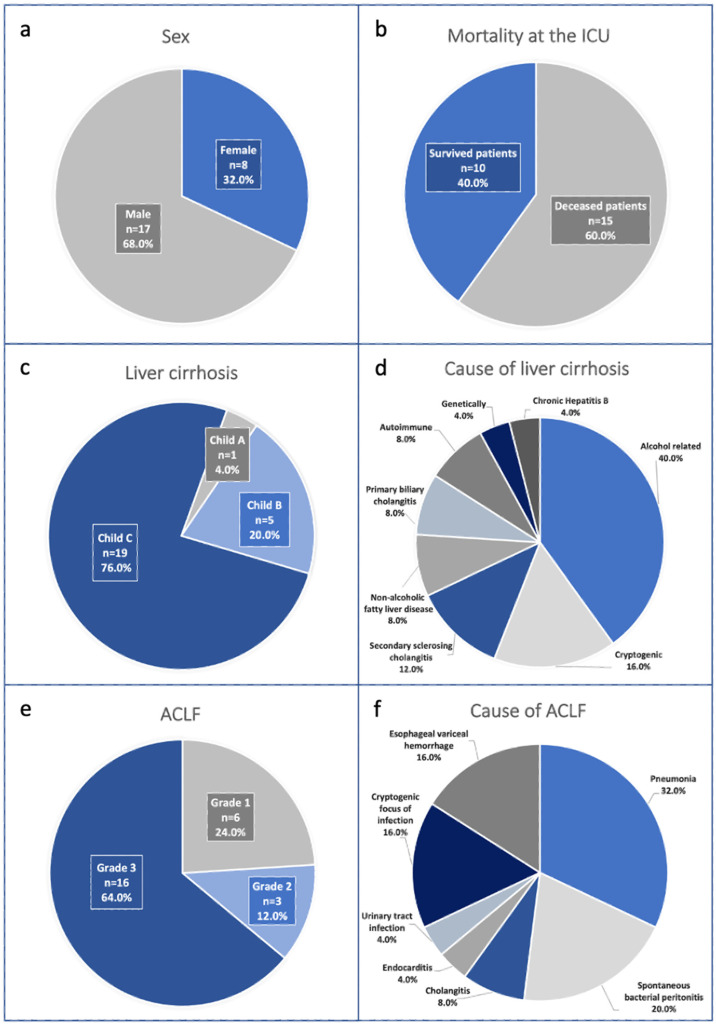
**Clinical characteristics of the study population.** In total, 25 patients with ACLF were included. All patients were diagnosed with liver cirrhosis and were treated at the ICU due to ACLF. (**a**): Distribution of sex; (**b**): Mortality in the MICU; (**c**): Liver cirrhosis measured by Child–Pugh score; (**d**): Etiology of liver cirrhosis; (**e**): ACLF Grade 1 to 3; (**f**): Causes of ACLF.

**Figure 2 antibiotics-12-01730-f002:**
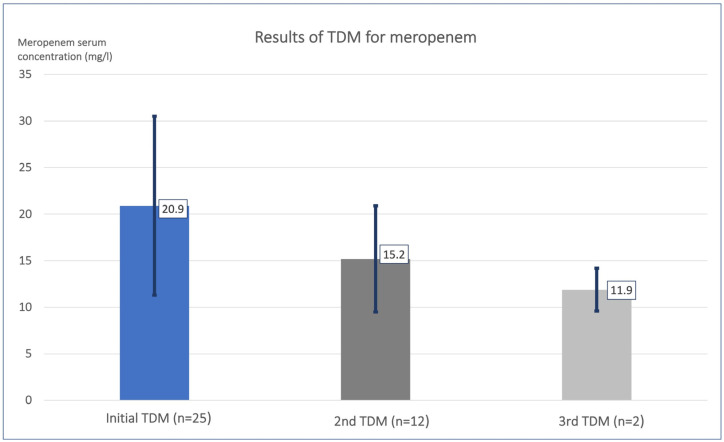
**TDM for meropenem with weekly follow-up.** Bar charts show the mean meropenem serum concentration in mg/L with standard deviation at the initial TDM, the second TDM in Week 2, and the third TDM in Week 3. Initial TDM for meropenem showed a mean meropenem serum concentration of 20.9 ± 9.6 mg/L. TDM = therapeutic drug monitoring.

**Figure 3 antibiotics-12-01730-f003:**
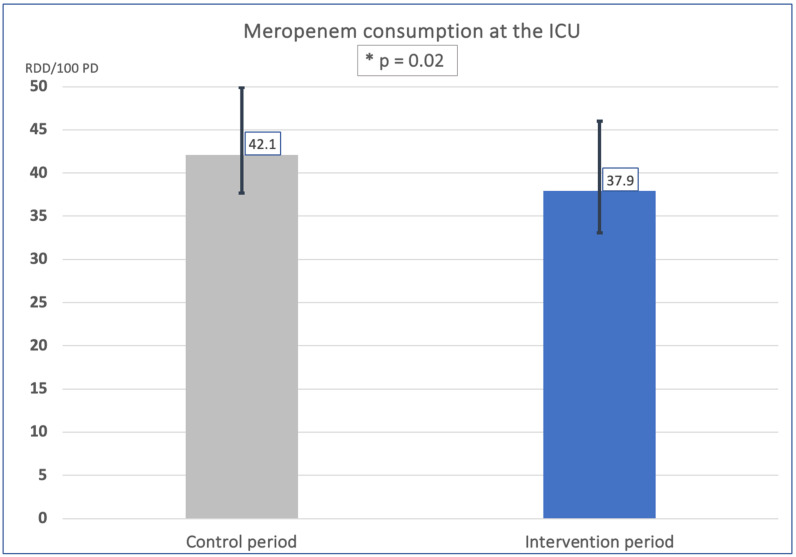
**Comparison of meropenem consumption in intervention and control period.** The diagram shows the mean meropenem application density and the range during the intervention period (Q4/2022–Q2/2023, 37.9 RDD/100 PD) and during the control period (Q4/2021–Q2/2022, 42.1 RDD/100 PD). There was a decrease of 10.0% in meropenem application density in the intervention period. RDD/100 PD = recommended daily doses (RDD) per 100 patient days (PD).

**Table 1 antibiotics-12-01730-t001:** Clinical characteristics of the study population.

Characteristics	Total Study Population (n = 25)
Age [years]: mean ± SD (range)	55.4 ± 13.90 (20–74)
Sex: n (%) Female Male	8 (32.0) 17 (68.0)
SOFA score [points]: mean ± SD (range)	15.8 ± 3.6 (9–22)
Mortality at the ICU: n (%) Deceased patients Survived patients	15 (60.0) 10 (40.0)
Liver cirrhosis: n (%) Child A/B/C	25 (100.0) 1 (4.0)/5 (20.0)/19 (76.0)
Cause of liver cirrhosis: n (%) Alcohol-related Cryptogenic Secondary sclerosing cholangitis Non-alcoholic fatty liver disease Autoimmune Primary biliary cholangitis Chronic Hepatitis B Genetic	10 (40.0) 4 (16.0) 3 (12.0) 2 (8.0) 2 (8.0) 2 (8.0) 1 (4.0) 1 (4.0)
MELD score [points]: mean ± SD (range)	27.9 ± 8.7 (14–40)
ACLF: n (%) ACLF Grade 1/2/3	25 (100.0) 6 (24.0)/3 (12.0)/16 (64.0)
Cause of ACLF: n (%) Pneumonia Spontaneous bacterial peritonitis Cholangitis Urinary tract infection Endocarditis Cryptogenic focus of infection Esophageal variceal hemorrhage	8 (32.0) 5 (20.0) 2 (8.0) 1 (4.0) 1 (4.0) 4 (16.0) 4 (16.0)
CLIF-C-ACLF score [points]: mean ± SD (range)	60.0 ± 9.3 (43–76)
Initial continuous dosing of meropenem [mg/h]: mean ± SD (range)	109 ± 20.8 (62.5–125)

Presentation of the baseline demographic (age, sex) and clinical (ICU stay, mortality in the ICU, liver cirrhosis, cause of liver cirrhosis, MELD score, ACLF, cause of ACLF, CLIF-C-ACLF score) characteristics of the total study population, and of the initial continuous dosing of meropenem.

**Table 2 antibiotics-12-01730-t002:** Results of TDM for meropenem.

Meropenem Serum Concentration (MSC)	Initial TDM (n = 25)	2nd TDM (n = 12)	3rd TDM (n = 2)
**MSC** [mg/L]: mean ± SD (range)	20.9 ± 9.6 (8.4–39)	15.2 ± 5.7 (9–24.6)	11.9 ± 2.3 (10.2–13.5)
**No. of MSCs inside target range (%)**	4 (16.0)	5 (41.7)	1 (50.0)
**No. of MSCs above target range (%)**	21 (84.0)	7 (58.3)	1 (50.0)
**No. of MSCs below target range (%)**	0 (0.0)	0 (0.0)	0 (0.0)

Presentation of the meropenem serum concentrations (MSC) collected during the initial, 2nd (Week 2), and 3rd (Week 3) TDM in our total study population.

**Table 3 antibiotics-12-01730-t003:** Recommendations of the interprofessional team.

Recommendations of the Interprofessional Team	Initial TDM (n = 25)	2nd TDM (n = 12)	3rd TDM (n = 2)
No change in meropenem dosage (%)	10 (40.0)	7 (58.3)	2 (100.0)
Decrease in meropenem dosage (%)	10 (40.0)	1 (8.3)	0 (0.0)
Increase in meropenem dosage (%)	0 (0.0)	0 (0.0)	0 (0.0)
Change to another antibiotic (%)	5 (20.0)	3 (25.0)	0 (0.0)
Stopping of antibiotic therapy (%)	0 (0.0)	1 (8.3)	0 (0.0)
Implementation (%)	100	100	100

Recommendations and implementation of the recommendations from the interprofessional team sorted by the initial, 2nd, and 3rd TDM.

**Table 4 antibiotics-12-01730-t004:** Meropenem application density at the ICU.

Meropenem Application Density (RDD/100 PD)	Q4	Q1	Q2
**Control period (Q4/2021–Q2/2022)**	49.9	39.4	37.7
**Intervention period (Q4/2022–Q2/2023)**	46.0	33.1	34.5

Meropenem application density in recommended daily doses (RDD)/100 patient days (PD) according to the control period and intervention period presented in the respective quarter of the year (Q).

## Data Availability

The raw data supporting the conclusions of this article will be made available by the authors, without undue reservation.

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
