# Peer review of "Interprofessional Therapeutic Drug Monitoring of Carbapenems Improves ICU Care and Guideline Adherence in Acute-on-Chronic Liver Failure"

_antibiotics, 2023, doi:10.3390/antibiotics12121730_

Round 1

Reviewer 1 Report

Comments and Suggestions for Authors

Dear Authors,

Very interesting article that sheds light on the importance of using TDM in establishing optimal therapeutic doses, especially in diseases that can endanger the patient's life. The complex interdisciplinary team in this study is remarkable, the pharmacokinetic and pharmacodynamic information associated with the use of carbapenems is also remarkable, especially in the conditions where there are bacterial species resistant to carbapenems, including expanded spectrum beta-lactamases (ESBL), AmpC cephalosporinases, and carbapenemases. Carbapenemase-producing Enterobacteriaceae (CRE) are of major importance for this class of antibiotics. The collaboration of different professionals from the hospital environment in this study offers an example of innovative activity that can be an example to follow, including the "I'm a-star" project initiated in this hospital unit.

The article is very well structured, informed and documented, it follows the general criteria of the journal, in my opinion, some very small adjustments are necessary to contribute to a clearer picture:

In section 2.3. the authors should also indicate the period (days) and the rhythmicity of the doses in which meropenem was administered. The same information should be added in section 4.2. – continuous administration of meropenem.

In section 3.3. - Results, in table 2 the authors should also specify the intervals in hours/days in which the results of Initial TDM, 2nd TDM and 3rd TDM were obtained, with continuous administration in which interval.

Reviewer 2 Report

Comments and Suggestions for Authors

The article provides the findings of a single center regarding the effects of implementation of interprofessional TDM for meropenem in ACLF patients. Few issues which need to be resolved are as mentioned below:

1.     The abstract is too lengthy and exceeds the word limit.

2.     Please mention full form of abbreviations in the abstract as well as text at their first appearance.

3.     There is duplicity of data represented in figures 1-3 and tables 1-3. I would suggest removing the figures.

4.     Under methods, please mention clearly the timing/s for blood sample collection for TDM with respect to drug administration.  For example, when were the sample/s taken for steady state and when were the subsequent samples taken.

5.     The authors have mentioned the results of TDM under 3 heads viz. initial TDM, 2nd and 3rd TDM. Please mention clearly what do these refer to? To my understanding these are the weekly TDM they have done, but nevertheless please specify in the text for better understanding of the readers.

6.     What was the rationale behind doing the three TDMs? What was the basis for weekly TDM?

7.     Since the study population comprised a small population of 25 patients, I would recommend adding a table depicting case-by-case clinical and demographic features, TDM parameters (steady state concentration, Css/MIC ratio) including the TDM team recommendations and final outcomes  etc. of all the patients to provide a better understanding and correlation to readers.

8.     For analysis of meropenem consumption in ICU, the authors have used the parameter: RDD/100 PDs. Please give reference for using this parameter as usually the consumption is defined in terms of Defined daily dose/ 100 bed days in inpatient settings. Also add briefly, how the consumption is calculated in this method.

9.     Regarding consumption, the authors mention 2 different terms viz. application density and consumption density. Please describe the difference between the two; if they refer to one parameter, it is better to use one to avoid any confusion.

10.  Figure 4: how was the mean application density and range calculated? Please provide the details.

11.   For the control period, need to add some details such as number of patients included, their clinical and demographic parameters etc. and any plasma drug level assessments of meropenem conducted to make a better comparison with the intervention period.

12.  Results from the study highlight the better results with TDM in terms of drug consumption levels. However, it will be interesting to appreciate the effect of interprofessional TDM versus control in terms of clinical outcomes per se. The same may be added under conclusions as well.

13.  Discussion point 4.5: The authors mention “TDM of meropenem promises substantial benefits for global health in both medium and long-term perspectives”; what perspectives are being referred to here? Also, in the absence of evaluation of such perspectives in the present study, they should not be discussed here. 

Comments on the Quality of English Language

Moderate editing of English language required.

Round 2

Reviewer 2 Report

Comments and Suggestions for Authors

The authors modified the manuscript as per the queries raised.